# Aquaculture Performance and Genetic Diversity of a New [(*Crassostrea hongkongensis* ♀ × *C*. *gigas* ♂) ♂ × *C. hongkongensis* ♀] Variety of the Oyster “South China No. 1” in Beibu Gulf, China

**DOI:** 10.3390/biology13050297

**Published:** 2024-04-26

**Authors:** Zonglu Wei, Yanping Qin, Haoxiang Liu, Qinggan Xing, Ziniu Yu, Yuehuan Zhang, Ying Pan

**Affiliations:** 1Key Laboratory of Aquatic Healthy Breeding and Nutrition Regulation of Guangxi Universities, College of Animal Science and Technology, Guangxi University, Nanning 530004, China; weizonglu@yeah.net (Z.W.); lhx1076190782@163.com (H.L.); 18289269632@163.com (Q.X.); 2CAS Key Laboratory of Tropical Marine Bio-Resources and Ecology, Guangdong Provincial Key Laboratory of Applied Marine Biology, South China Sea Institute of Oceanology, Innovation Academy of South China Sea Ecology and Environmental Engineering, Chinese Academy of Sciences, Guangzhou 510301, China; yanpingqin@scsio.ac.cn (Y.Q.); carlzyu@scsio.ac.cn (Z.Y.); yhzhang@scsio.ac.cn (Y.Z.); 3Southern Marine Science and Engineering Guangdong Laboratory (Guangzhou), Guangzhou 511458, China; 4Hainan Provincial Key Laboratory of Tropical Marine Biology Technology, Sanya Marine Eco-Environment Engineering Research Institute, Tropical Marine Biological Research Station in Hainan, Chinese Academy of Sciences, Sanya 572024, China

**Keywords:** oyster “South China No. 1”, *Crassostrea hongkongensis*, stocking density, genetic diversity, Beibu Gulf

## Abstract

**Simple Summary:**

A new interspecific backcross ([(*Crassostrea hongkongensis* ♀ × *C. gigas* ♂) ♂ × *C. hongkongensis* ♀]) variety was bred by the South China Sea Institute of Oceanology and named “South China No. 1”. This study aims to explore the effects of stocking density on the growth performance of “South China No. 1”, and compare the growth performance and genetic diversity to that of *C. hongkongensis* in Beibu Gulf. The results showed that oysters bred under a stocking density of 20 oysters/substrate had better growth manifestation. It was found that the shell height and total weight of “South China No. 1” cultured in Fangchenggang were significantly higher than that of those in Beihai and Qinzhou from September 2018 to November 2018. In addition, the shell height and shell width of “South China No. 1” were significantly higher than that of *C. hongkongensis*. In the Hardy–Weinberg equilibrium test, for the seven populations and ten microsatellites, in 10 of the 70 groups, the segregation distortion was significant. These results suggest that a stocking density of 20 oysters/substrate can promote the shell height, shell width and total weight of “South China No. 1”. Fangchenggang is a suitable place to cultivate the “South China No. 1” breed according to the total weight and sum of all algal genus abundances.

**Abstract:**

*Crassostrea hongkongensis* is an economically important bivalve found in various parts of the South China Sea. A new interspecific backcross ([(*Crassostrea hongkongensis* ♀ × *C*. *gigas* ♂) ♂ × *C. hongkongensis* ♀]) variety was bred by the South China Sea Institute of Oceanology which named “South China No. 1”. This study aims to explore the effects of stocking density on the growth performance of “South China No. 1”, compared their growth performance and genetic diversity to *C. hongkongensis*, and found the best place breeding site for “South China No. 1” in Beibu Gulf. The results showed that stocking a density of 20 oysters/substrate can significantly increase the shell height, shell width, total weight, survival rate, daily shell height gain and daily body mass gain. It was found that the shell height and total weight of “South China No. 1” cultured in Fangchenggang were significantly higher than that of those in Beihai and Qinzhou from September 2018 to November 2018. Similarly, the shell width of oysters in Fangchenggang and Qinzhou was also significantly higher in September 2018, and the interaction between site and stocking density had significant effects on the shell width in March 2018 and November 2018. In addition, the shell height and shell width of “South China No. 1” were significantly higher than that of *C. hongkongensis* in all three sites. At all three sites, the phytoplankton community structure was mostly dominated by Bacillariophyta. In the Hardy–Weinberg equilibrium test, for the seven populations and ten microsatellites, in 10 of the 70 groups, the segregation distortion was significant. These results suggest that a stocking density of 20 oysters/substrate can promote the shell height, shell width and total weight of “South China No. 1” in Beibu Gulf, China. “South China No. 1” has better growth performance compared with *C. hongkongensis*. Fangchenggang is a suitable place to cultivate the “South China No. 1” breed according to the total weight and sum of all algal genus abundances. The results of this study can be used as a reference to further understand the stocking density and genetic diversity of the “South China No. 1” breed in Beibu Gulf, China.

## 1. Introduction

Oyster is an important mollusk, which is a significant source of nutrients for human [1,2]. The world’s oyster aquaculture production is approximately 6.06 million tons, accounting for 31.4% of the total global mollusk production in 2020 [3]. In 2019, the total production of oyster accounted for 36.31% (5.23 million tons) of total mollusk aquaculture production [4], while *Crassostrea hongkongensis* accounted for 37.43% (2.0 million tons) of total oyster production in China [5]. The Beibu Gulf is a crucial mollusk aquaculture area, which is located at the northwest of the South China Sea, especially in Qinzhou, Fangchenggang and Zhenzhu bays. [6,7]. Since 2019, the second largest *C. hongkongensis* aquaculture farm in China is in Qinzhou, accounting for 32.67% of total oyster production, and contributing 5700 million CNY [5].

*C. hongkongensis* is deemed one of the most important bivalves in the South China Sea. The species formerly collectively known as “*C. ariakinsis*” was first named as *C. hongkongensis* in 2004 [8]. In 2006, our team conducted a cross-breeding study on *C. hongkongensis* by using mixed selection and molecular identification. In 2016, a new variety of [(*C*. *hongkongensis* ♀ × *C*. *gigas* ♂) ♂ × *C*. *hongkongensis* ♀], namely “South China No. 1”, was successfully bred [9]. The parental base group of “South China No. 1” breeding is the backcross generation of (*C*. *hongkongensis* ♀ × *C*. *gigas* ♂) ♂ and *C*. *hongkongensis* ♀, which has a strong growth advantage combination [9]. Total weight and yield of “South China No. 1” was 17.1% and 23.1% higher than that of ordinary *C. hongkongensis* [9]. Studies have shown that the optimum salinity for the growth of planktic larvae and larvae in “South China No. 1” was 8 to 28, and the optimum salinity was 13 to 23, which expands the existing farming area [10].

Stocking density is one of the elements affecting the growth and survival of oyster [11,12,13,14]. High stocking density leads to a reduction in the growth of oysters in the nursery and grow-out phases [15,16], which may be related to space and food constraints [17]. Bivalves that feed on phytoplankton, zooplankton and suspended solids have a great influence on their survival and growth [18]. Microsatellite markers have the advantages of large quantity, wide distribution and uniformity, high polymorphism information content, co-dominant inheritance, and convenient analysis, and have been extensively applied in the genetic breeding of aquatic animals [19]. There have been many studies on the genetic diversity of artificially selected populations of some economic shellfish, such as *C. gigas* [20], *Crassostrea sikamea* [21] and *Crassostrea virginica* [22]. Information on the genetic diversity of “South China No. 1” is still limited. Therefore, the purpose of this study is to explore the effects of stocking density and location on the grow-out traits of “South China No. 1”. In addition, we also compared the grow-out traits and genetic diversity of “South China No. 1” and *C. hongkongensis* to better evaluate the growth performance of “South China No. 1”. The results of this study may have certain guiding significance for the breeding and aquaculture of “South China No. 1” in Beibu Gulf, China. It is helpful to further improve and promote the production and development of oyster backcross varieties.

## 2. Materials and Methods

### 2.1. Experimental Animals and Study Area

The “South China No. 1” breed was bred through backcross of (*Crassostrea hongkongensis* ♀ × *C*. *gigas* ♂) ♂ and *C*. *hongkongensis* ♀. The “South China No. 1” breed and the control group of *C. hongkongensis* were fertilized at the same time and bred in the Zhanjiang Marine Economic Animal Experiment Station of the South China Sea Institute of Oceanography. After a period of time, they were collected at the same time and transferred to the experimental site. The experimental sites were located in three sea areas along the coast of the Beibu Gulf in China, namely Beihai (21°55′N, 109°46′ E), Qinzhou (21°40′ N, 108°42′ E) and Fangchenggang (21°31′ N, 108°14′ E) (Figure 1). The temperature, salinity and pH were 18–28 °C, 16–18 ppt and 7.6–8.4, respectively. Experiments were conducted from January 2018 to January 2019.

### 2.2. Growth Experiment

A total of 540 (“South China No. 1”) and 540 (*C. hongkongensis*) juvenile oysters were separately kept in 6 lanterns which was a cylindrical structure with a mesh opening of 0.8 cm, had 6 substrates, and was 15 cm high and 30 cm in diameter [23]; each substrate had a density of 20, 30 and 40 oysters/substrate, respectively, and was sent to the experimental breeding site at the same time. Every two mouths, all oysters were counted, weighed and the shell height, shell length and shell width were measured to calculate the daily shell height gain and the daily body mass gain, then continued breeding until the end of the experiment. One extra lantern with the same density as the experimental group was used to replace the dead oysters [24]. The mean initial shell height, shell length, shell width and total weight of “South China No. 1” and *C. hongkongensis* were 63.90 ± 9.29 mm and 62.73 ± 7.07 mm, 45.86 ± 7.00 mm and 44.89 ± 7.93 mm, 23.13 ± 3.05 mm and 21.04 ± 3.64 mm and 43.11 ± 7.23 g and 43.52 ± 14.41 g, respectively. 

### 2.3. Environmental Parameters of Seawater and Phytoplankton Data

A US YSI portable water quality analyzer (Procomm II) was used to analyze the pH and dissolved oxygen in seawater. Phytoplankton are identified according to the criteria of Muthukumaravel et al. [25]. Every two mouths, 1 L of seawater sample was collected between two breeding lanterns at seawater depth, approximately 90 cm, to inform the three study sea areas, used to authenticate phytoplankton. The collected 1 L water sample was fixed with formaldehyde (CH_2_O), and after the algae sample was settled for 24 h to 48 h by the algae settler, the 1 mL sample was transferred to the phytoplankton counting chamber (0.1 mL, 20 mm × 20 mm), and the algae was identified as a genus. Follow-up observations were made under the Nikon Research Inverted Light microscope (C-W15x).

### 2.4. The Genetic Diversity of 7 Cultured Populations

Over the 360-day experiment, two oysters were randomly collected from each substrate at three sites. In addition, an equal number of *C. gigas* were randomly collected from Qingdao in July 2019, and its shell height, shell length, shell width and total weight were 103.38 ± 11.43 cm, 55.66 ± 6.47 cm, 34.04 ± 4.69 cm and 90.27 ± 17.73 g, respectively. For the genetic diversity experiment, the adductor muscle tissue of oysters was extracted, immersed in absolute ethanol, and then stored at −80 °C.

### 2.5. DNA Isolation and PCR Extension

The total DNA extraction method was described by Zhang et al. [26]. NanoDrop (Thermo Fisher Scientific, Sunnyvale, CA, USA) was used to detect the purity and concentration of DNA. Agarose gel (1.2%) (Trans Gen, Beijing, China) was used to confirm the integrity of DNA. Primers were cited from Xiao et al. [27] and Ma et al. [28]. All the primers in Table 1 were synthesized by Shanghai Sangon Biotech Co. Ltd. (Shanghai, China). The total reaction volume was 10 μL, containing 1 µL DNA, 1 μL forward primer (10 μM) and 1 μL reverse primer (10 μM), 2 µL ddH_2_O, and 5 µL Taq Mix (Trans Gen, Beijing, China). The PCR cycle program was: 1 cycle at 95 °C for 3 min, then 35 cycles at 95 °C for 45 s, 56 °C for 45 s, 72 °C for 45 s, and then 1 cycle at 72 °C for 5 min. Non-denaturing polyacrylamide gel (8%) was used for electrophoresis detection of the product of the PCR extension, compared with the 10 bp DNA ladder. After silver nitrate staining, the results were screened with a gel scanner (Bio-Rad, Hercules, CA, USA).

### 2.6. Statistical Analysis

Normality and mean-square error assumptions were confirmed before any statistical analysis. A two-factor Analysis of Variance (ANOVA) was utilized to establish the effects of location, stocking density and their interaction on growth, total weight, survival rate, daily shell height gain and daily body mass gain. Logarithmic transformation was performed on growth data before analysis, arcsine transformation was performed on incremental survival rate data before analysis, and bidirectional statistical significance was evaluated [29]. One-way Analysis of Variance fand Duncan’s multiple comparison test was used to compare growth parameters of the two breeds with different stocking densities. In addition, a subsequent trend analysis was performed using orthogonal polynomial contrasts to determine whether the significant effects were linear and/or quadratic. The main effect and interaction of two factors (stocking density and breed) were analyzed by two-factor Analysis of Variance. Data were expressed as the mean ± standard deviation. *p* < 0.05 was considered statistically significant. All analyses were performed using SPSS Statistics 26.0 (SPSS, Michigan Avenue, Chicago, IL, USA). 

Species with a degree of dominance in the microalgae community greater than 0.2 are defined as dominant species.

Dominance [30]
$$Y=\frac{n_{i}}{N\times f_{i}}$$

Y represents dominance, N represents the total cell abundance of the community, n_i_ represents the cell abundance of No. i species in the community, and f_i_ represents the frequency of the No. i species occurring at each station in the community.

Electrophoretic band analysis was performed using Quantity One, and the size of bands was estimated after manual calibration. The number of alleles (*N_A_*), observed heterozygosity (*H_O_*) and expected heterozygosity (*H_E_*) were calculated by Popgen 32.0. The polymorphism information content was calculated by Cervus 3.0. The polymorphic information content was calculated by Gervus, and Hardy–Weinberg’s equilibrium was calculated for every locus population by Genepop though the allele frequency. According to the genetic distance between populations, a clustering relational tree was determined for every aquaculture populations by MEGA 7.0. 

## 3. Result

### 3.1. Effect of Stocking Density and Site on the Growth of “South China No. 1”

Overall, from March 2018 to November 2018, the shell height, shell width and total weight of “South China No. 1” increased continuously at all three sites and three stocking densities (Table 2). Site and stocking density had a significant impact on shell height, shell width and total weight from March 2018 to November 2018 (*p* < 0.05). The total weight was 72.47 g, 73.02 g and 81.88 g at Beihai, 66.42 g, 79.64 g and 94.28 g at Qinzhou, and 69.13 g, 89.41 g and 98.41 g at Fangchenggang in July 2018, September 2018 and November 2018, and had the highest value at Fangchenggang from September 2018 to November 2018 (*p* < 0.05). In September 2018 and November 2018, the shell height and shell width of “South China No. 1” and farmed in Fangchenggang were significantly higher than that of those farmed in Beihai and Qinzhou (*p* < 0.05). With the increase in stocking density, the shell height, shell width and total weight of oysters decreased significantly (*p* < 0.05). At all stations, the highest shell height, shell width and total weight of oysters were recorded when stocking density was 20 oysters/substrate (*p* < 0.05). The interaction between site and stocking density had significant effects on shell width in March 2018 and November 2018 (*p* < 0.05).

In March 2018 to May 2018, there was a significant difference in the survival rate (SR) of oysters at all three sites, with oysters farmed in Fangchenggang having the highest SR (*p* < 0.05). The stocking density significantly affected SR and daily body mass gain (DMG) (*p* < 0.05), but did not affect the daily shell height gain (DHG) in September 2018 (*p* > 0.05). There was no significant difference in the site and stocking density of DHG (*p* > 0.05), except for in March 2018 (*p* < 0.05). In March 2018, May 2018, and September 2018, the DMG of oysters with a stocking density of 20 oysters/substrate was higher than that of those with a stocking density of 40 oysters/substrate (*p* < 0.05). However, the interaction between site and stocking density has no significant impact on SR, DMG and DHG (*p* > 0.05) (Table 2).

### 3.2. Effect of Stocking Density on the Growth of Oyster “South China No. 1” and C. hongkongensis Conducted for 360 Days

In Beihai and Qinzhou, the shell height, shell length, shell width and total weight of “South China No. 1” with a stocking density of 20 oysters/substrate was significantly higher than that of those with other stocking densities (Table 3 and Table 4) (*p* < 0.05). At all three sites, compared to *C. hongkongensis*, “South China No. 1” showed significantly higher shell height, shell length, shell width and total weight (Table 3, Table 4 and Table 5) (*p* < 0.05). In Beihai, the interaction between breed and stocking density has significant effects on shell height (Table 3). In Qinzhou, the interaction between breed and stocking density has significant effects on shell width (Table 4). In Fangchenggang, the interaction between breed and stocking density has significant effects on total weight (Table 5) (*p* < 0.05). 

### 3.3. Seawater Quality and Plankton at Three Sites

The environmental factors of the seawater temperature, salinity, pH and dissolved oxygen were shown in Figure 2. No significant differences in environment factors including temperature, salinity, pH and dissolved oxygen at three sites were found. Seawater temperatures ranged from 14 °C to 31 °C at Beihai, from 15 °C to 30 °C at Qinzhou and from 15 °C to 32 °C at Fangchenggang. Seawater salinity ranged from 16 ppt to 24 ppt at Beihai, from 16 ppt to 26 ppt at Qinzhou and from 14 ppt to 27 ppt at Fangchenggang. pH ranged from 7.62 to 8.4 at Beihai, from 7.84 to 8.44 at Qinzhou and from 7.82 to 8.3 at Fangchenggang. However, from September 2018 to November 2018, seawater salinity in different locations varied greatly, with the salinity of the sea water in Beihai ranging from 16.5 to 18 ppt, salinity of the sea water in Qinzhou ranging from 16.5 to 22.4 ppt, and the salinity of the sea water in Fangchenggang ranging from 24 to 26.7 ppt.

As shown in Table 6, *Skeletonema* was the most abundant algal species, the only dominant species, which the abundance was 4193 cells·L^−1^, 4589 cells·L^−1^ and 6343 cells·L^−1^ in Beihai, Qinzhou and Fangchenggang, respectively. In Beihai, Qinzhou and Fangchenggang, the sum of all algal genus abundances was 5928 cells·L^−1^, 6466 cells·L^−1^ and 9118 cells·L^−1^, respectively. *Skeletonema*, *Pleurosigma* and *Navicula* had the highest frequency at all three sites. The lowest algal abundance was recorded in *Triceratium* in Beihai and Fangchenggang, and in *Synedra* in Qinzhou. *Skeletonema*, *Pleurosigma*, *Eucampia*, *Navicula*, *Ditylum*, *Chaetoceros*, *Rhizosolenia*, *Thalassionema*, *Coscinodiscus*, *Synedra*, *Nitzschia, Bacillaria*, *Licmophora*, *Thalassiosira* and *Lauderia* were found at all three sites. *Triceratium* was found in Beihai and Fangchenggang. *Cerataulina* was found in Beihai and Qinzhou. *Biddulphia* and *Odontella* were found in Qinzhou and Fangchenggang. 

### 3.4. The Genetic Diversity of Seven Cultured Populations

As shown in Table 7, the average allele number for 10 pairs of microsatellite primers in 7 oyster cultured population ranged from 4.9 to 5.7. The average number of effective allele, average observed heterozygosity and expected heterozygosity ranged from 2.29 to 3.03, 0.61 to 0.83 and 0.54 to 0.67, respectively. *C. hongkongensis* from Beihai was first clustered with *C. hongkongensis* from Fangchenggang, and then with *C. hongkongensis* from Qinzhou. The “South China No. 1” breed from Beihai was first clustered with “South China No. 1” from Qinzhou, and then clustered with “South China No. 1” from Fangchenggang, while the *C*. *gigas* from Qingdao belonged to the outermost branch (Figure 3).

## 4. Discussion

Stocking density has a significant effect on the growth traits and survival rate of bivalves. Our study found that the shell height, shell width and total weight of “South China No. 1” were significantly affected by stocking density, and the best stocking density was 20 oysters/substrate. The highest shell length and total weight were obtained when “South China No. 1” and *Crassostrea hongkongensis* were bred in a stocking density of 20 oysters/substrate at Beihai and Qinzhou. It has been reported that *C. gigas* and *C. angulata* at 20 oysters/basket generally outperformed those at 30 and 40 oysters/basket, which has comparability to the results of our research [12]. A report showed that the growth of *C. gigas* cultivated at a stocking density of 50 larvae mL^−1^ was significantly higher than that of those cultivated at stocking densities of 75 and 100 larvae mL^−1^ [11]. Previous studies have also reported that a low stocking density has better growth traits for abalone [31], Chinese tiger frog (*Hoplobatrachus rugulosa*) [32] and sea cucumber (*Apostichopus japonicus*) [33]. Similarly, as the stocking density increases, the growth traits of aquatic animals, such as fish [34] and shrimp [35], also decrease. The factors that affect the growth of organisms in high-density environments may be competition for food and space [15,36,37]. In this study, in May 2018 and September 2018, the survival rate of “South China No. 1” cultivated at 20 oysters/substrate was significantly higher than that at 30 and 40 oysters/substrate. Oliveira et al. [24] found that compared to at a higher stocking density, the survival rates of post-larvae at lower densities were significantly higher. Yang et al. [38] revealed that stocking density significantly affects growth and survival of dwarf surfclam (*Mulinia lateralis*) larvae. The superior stocking density of D-shaped larvae, umbo larvae and pediveliger larvae was ≤9 larvae mL^−1^, ≤7 larvae mL^−1^ and ≤5 larvae mL^−1^, respectively. These observations support our results that survival is connected with the stocking density of “South China No. 1”. Within a certain density range, mortality is not affected by stocking density; when the density exceeds the threshold, mortality rate will increase [37,39]. 

As one of the genetic improvement methods in breeding, interspecific backcross plays an important role in improving the performance of economically important strains [40,41,42]. In the present study, we found that breed has a great influence on growth, in which the shell height and shell width of “South China No. 1” at all three sites were significantly higher than that of the common *C. hongkongensis*. Moreover, the differences were more obvious when oysters were bred in Fangchenggang. The shell height, shell length, shell width and total weight of “South China No. 1” were higher than that of those of *C. hongkongensis*, indicating that the backcross varieties had certain growth advantages. Zhang et al. [41] demonstrated that artificial interspecific mating between *C. hongkongensis*♀ × *C. gigas*♂ fertile hybrids and their parents could successfully produce backcross offspring with a high fertilization level, a fast growth rate and a high survival rate. Research has shown that backcross populations between the hybrid *Haliotis discus hannai* ♀ × *H. fulgens* ♂ and the parent species had significant advantages in growth and survival [43]. Shi et al. [44] found that there were differences in the growth of backcrossed populations between the *C. angulata* ♀ × *C. gigas* ♂ hybrids and their parents. Similar observations have been documented in fish [38,42,45]. For example, Lago et al. [45] found that the development of genetic strains of red tilapia could make use of genetic breeding programs, mainly to explore paternal heterosis to produce fish. Researchers utilized artificial insemination technology to generate a backcross variety (BC) [*Megalobrama amblycephala* (MA) ♀ × (*Megalobrama amblycephala* ♀ × *Culter alburnus* (CA) ♂) ♂], which has superior growth advantage and digestive enzyme activity than that of its parents [42]. The study on backcross grouper showed that its growth rate was better than of its female parent; the growth difference increased with age [38]. These observations support our results that the backcross offspring have stronger growth advantages than the parents. 

Environmental factors, especially salinity, have a significant impact on aquatic animals [46]. Bivalves are filter-feeders and thereby remove the associated nutrient particles from the water column, so their growth performance and feeding depend on environmental factors, like salinity [47]. Our study found that in September 2018 to November 2018, seawater salinity was more stable and higher in Fangchenggang compared with in Beihai and Qinzhou. Furthermore, in September to November 2018, the growth indicators of the shell height, shell width and total weight of “South China No. 1” in Fangchenggang were significantly higher than that of those in Beihai. The research of salinity on early development and growth “South China No. 1” showed that suitable salinity for the larvae and spat of “South China No. 1” ranged from 13 to 23 [10]. The salinity of “South China No. 1” in different growth stages varies, and when fattening, it should be transferred to a high salt environment of 20 to 25 ppt [9]. When oysters are in the juvenile stage, their resistance to hypersaline environment is low [48]. As the oyster grows, it becomes more resistant to high salt levels [48]. Peng et al. [49] found that the shell length and weight gain rate of *Sinonovacula constricta* were significantly lower when it was bred in the low-salinity environment during the first month of the experiment. Similarly, *Pecten maximus* at 20 ppt had a significantly higher mortality than that of those at 25 and 30 ppt [50]. Liu et al. [51] investigated the influences of five different salinities (20, 25, 30, 35 and 40 ppt) on growth, tissue damage, enzymes and body composition of *Babylonia areolata* after salinity adaptation for seven weeks. They found that when juvenile ivory shells were bred in 35 and 40 ppt, growth and survival will be inhibited. The fluctuation in salinity affects the osmolality of aquatic animals, resulting in physiological and metabolic changes, tissue damage, and irregular growth [52]. Therefore, stable salinity could provide a better growing environment for bivalves, and this maybe the reason that “South China No. 1” had better growth performance in Fangchenggang in September 2018 to November 2018. 

As we know, low salinity could affect the growth, reproduction and photosynthesis of some algae [53,54]. Bivalves feed on many algae, such as phytoplankton, resuspended benthic microalgae and detritus [47]. Phytoplankton are a food source for bivalves and help to maintain the primary productivity of fisheries [55]. To a extent, mariculture animal’s yield and quality determine that of phytoplankton [56]. The phytoplankton community structure investigated in our study was dominated by diatomata, which is basically consistent with the phytoplankton community structure reported in previous studies [47,57,58]. Bacillariophyta is the preferred food of many herbivores and is less toxic, so it is considered better for the marine ecosystem. [30]. The high biomass of diatoms was related to the bloom of *Skeletonema*. It has been reported that *Skeletonema* prevails in warmer climates [59]. Temperature has an important effect on the variation in Bacillariophyta [58]. In Fangchenggang, the abundance of the phytoplankton community was higher than in Beihai and Qinzhou, providing more food for oysters. Maybe this is one of the reasons why “South China No. 1” had better growth performance in Fangchenggang. 

Allelic genetic diversity and heterozygosity can be used as indicators of population genetic structure variation [60]. Compared with the average observed heterozygosity and expected heterozygosity of the seven populations, the heterozygosity of the selected population was slightly increased. The mean effective allele number (*N_E_*) and mean expected heterozygosity (*H_E_*) of “South China No. 1” in the three cultured sea sites were slightly lower than that of *C. hongkongensis* in the same sea site, indicating a decrease in the genetic diversity of the selected population during the continuous artificial breeding process. In this study, we found that “South China No. 1” had a distant relationship with *C. gigas* of Qingdao, and a closer genetic relationship with *C. hongkongensis*. This may be attributed to the selection of primers from the developed *C. hongkongensis* microsatellite primers. In addition, due to a small number of parents unintentionally selecting the best offspring, the selection of successive generations accelerates the homogenization of germplasm resources [61], and also leads to a reduction in population genetic diversity. At present, in many aquatic economic animals, such as *Lates calcarifer* [62], silver-lipped pearl oysters (*Pinctada maxima*) [63] and rock oyster (*Saccostrea glomerata*) [64], a certain degree of genetic diversity reduction has been observed during artificial selection. The results of this study showed that after Bonferroni correction, bias separation occurred in all seven populations. The 10 sites, except Ch423, Ch697, Ch307 and Ch411, did not deviate from the Hardy–Weinberg equilibrium for seven populations, and bias separation occurred in the other six sites. This may be because the invalid alleles cause changes in the gene frequency, which further leads to the Hardy–Weinberg imbalance, similar to the results of the study of Reece et al. [65], which found that many shellfish have invalid alleles.

## 5. Conclusions

In summary, stocking density is a vital element affecting growth performance, in which lower stocking density (20 oysters/substrate) significantly increased the shell height, shell width, total weight, survival rate, daily shell height gain and daily body mass gain of “South China No. 1” in Beibu Gulf, China. In Beihai, Qinzhou and Fangchenggang, the shell height and shell width of “South China No. 1” were higher than that of *Crassostrea hongkongensis*, indicating that “South China No. 1” had better growth performance. The main algal genus in Beibu Gulf, China was Bacillariophyta. In Beihai, Qinzhou and Fangchenggang, the sum of all algal genus abundances was 5928 cells·L^−1^, 6466 cells·L^−1^ and 9118 cells·L^−1^, respectively. The “South China No. 1” breed is closely related to *C. hongkongensis*, and has high genetic variability. Based on the total weight and sum of all algal genus abundances, the better place for culture “South China No. 1” is Fangchenggang.

## Figures and Tables

**Figure 1 biology-13-00297-f001:**
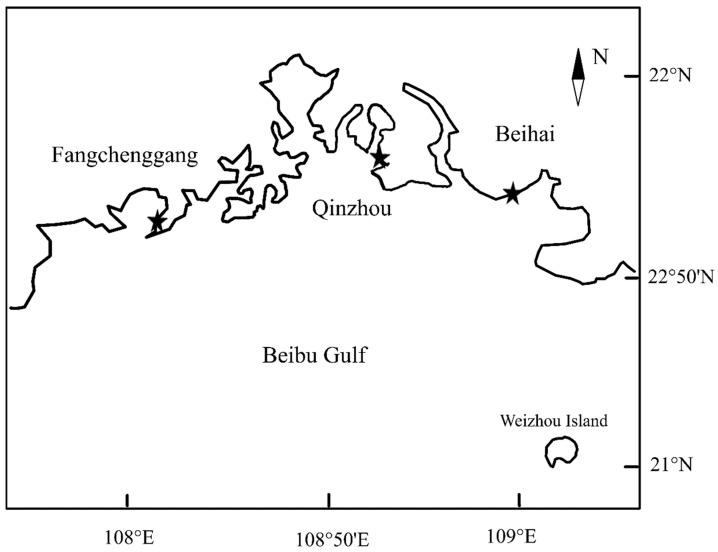
Location of the experimental grow-out sites in Beibu Gulf, China (* = Beihai, Qinzhou and Fangchenggang).

**Figure 2 biology-13-00297-f002:**
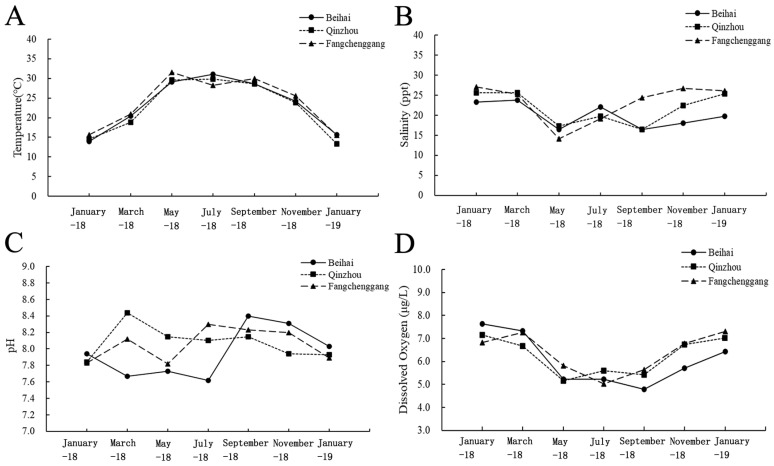
The variations in (**A**) temperature, (**B**) salinity, (**C**) pH and (**D**) dissolved oxygen at the Beihai, Qinzhou and Fangchenggang sites from January 2018 to January 2019.

**Figure 3 biology-13-00297-f003:**
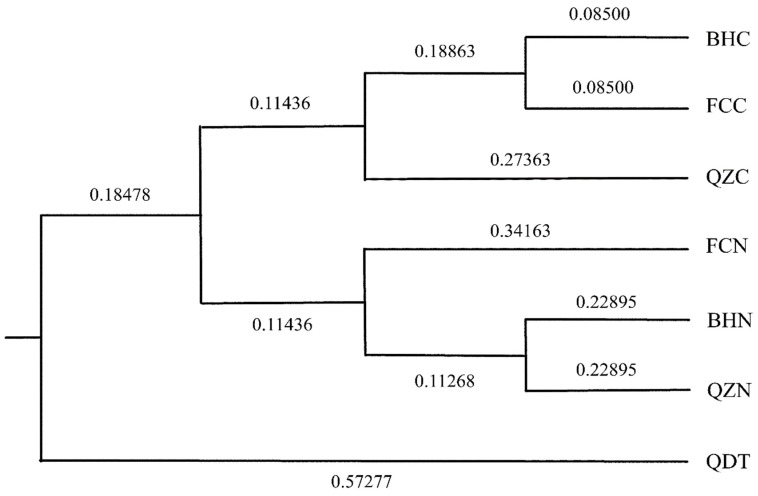
The UPGMA cluster analysis chart of seven oyster cultured populations based on Nei’s genetic unbiased genetic distance.

**Table 1 biology-13-00297-t001:** Information of ten pairs of microsatellite primers.

Locus	Annealing Temperature	Size	Repeat Motif Sequence	Primer Sequence (5′–3′)
Ch311	60	208–282	(GT)_n_(GTGC)_n_	F: GGACGAAATGGAAAGTGTA R: CGCGTTTGCCAATAACCT
Ch319	60	408–456	(CA)_n_	F: GTCGCACAATGAGTAAAGCA R: AAGAGGGTGGGTGGAGTA
Ch423	60	222–374	(GGT)_n_ (GAT)_n_	F: ACCGTCGTTGTCGTCTCA R: CGTCCTCAGGTCACTTTC
Ch417	60	132–168	(GAT)_n_	F: GTGAGTGCGGTGGTTTCT R: CTACCTTCTGTGCTGGATGA
Ch405	60	252–279	(GAT)_n_	F: AGAGGTCGTGTTAGAGATGG R: AAGATGATACTGCTATGGAAG
Ch697	60	168	(TGT)_n_(TTG)_n_	F: CTGTTGAGCCAGTTCCATGA R: GGACAATACGGTCAGCAACA
Ch307	50	244–298	(CA)_n_	F: AACCCATCCGCAAACAAT R: ATCCAACTGAACACCACCAT
Ch317	50	201–261	(CA)_n_	F: CATTGCCGCACCCATTTA R: TTCCGGTCTATCTTCTGA
Ch411	60	208–244	(GAT)_n_	F: CCGCCAGTGTCATCCTCA R: CCAGCAGGGCTTTAGACG
Ch414	60	236–273	(GAT)_n_	F: CAATGATGTAGAGGTCGT R: GATGATACTGCTATGGAAGA

Note: F = the forward primer; R = the reverse primer.

**Table 2 biology-13-00297-t002:** Effects of site, stocking density and their interaction on shell height, shell width, total weight, survival rate, daily shell height gain and daily body mass gain of oyster “South China No. 1” from March 2018 to November 2018.

			Shell Height (mm)		Shell Width (mm)		Total Weight (g)		SR (%)		DHG (mm/d)		DMG (g/d)	
			Means	*p*-Value	Means	*p*-Value	Means	*p*-Value	Means	*p*-Value	Means	*p*-Value	Means	*p*-Value
Time	Treatments													
March 2018 (60th day)	site	Beihai	71.71 ^A^	0.029	24.65 ^A^	0.003	50.84 ^A^	0.004	92.46 ^B^	0.007	0.13 ^A^	0.042	0.13 ^A^	0.010
		Qinzhou	71.11 ^A^		24.49 ^A^		52.19 ^A^		97.88 ^A^		0.12 ^AB^		0.15 ^A^	
		Fangchenggang	69.66 ^B^		23.39 ^B^		48.13 ^B^		97.70 ^A^		0.10 ^B^		0.08 ^B^	
	stocking density	20	72.07 ^A^	0.035	25.32 ^A^	<0.001	51.71 ^A^	0.002	95.00	0.622	0.14 ^A^	0.034	0.14 ^A^	0.020
		30	70.35 ^B^		23.47 ^B^		51.25 ^A^		95.92		0.11 ^B^		0.14 ^A^	
		40	70.05 ^B^		23.74 ^B^		48.20 ^B^		97.11		0.10 ^B^		0.08 ^B^	
	site × stocking density			0.658		0.004		0.296		0.564		0.692		0.367
May 2018 (120th day)	site	Beihai	74.73 ^A^	<0.001	26.05 ^A^	<0.001	64.58 ^A^	0.024	99.02 ^A^	0.008	0.05	0.493	0.23	0.302
		Qinzhou	73.55 ^A^		26.79 ^A^		62.91 ^AB^		95.38 ^B^		0.04		0.18	
		Fangchenggang	71.39 ^B^		24.59 ^B^		60.55 ^B^		98.99 ^A^		0.03		0.21	
	stocking density	20	75.86 ^A^	<0.001	26.60 ^A^	<0.001	69.22 ^A^	<0.001	98.33 ^A^	0.003	0.06	0.087	0.29 ^A^	<0.001
		30	72.03 ^B^		25.07 ^B^		60.97 ^B^		97.39 ^B^		0.03		0.16 ^B^	
		40	71.78 ^B^		25.76 ^B^		57.86 ^C^		97.67 ^B^		0.03		0.16 ^B^	
	site × stocking density			0.281		0.061		0.210		0.117		0.432		0.588
July 2018 (180th day)	site	Beihai	74.95 ^A^	<0.001	27.32 ^B^	0.016	72.47 ^A^	0.002	99.40	0.070	0.01	0.267	0.13 ^A^	0.037
		Qinzhou	74.24 ^B^		27.03 ^B^		66.42 ^B^		98.30		0.01		0.07 ^B^	
		Fangchenggang	74.85 ^A^		28.07 ^A^		69.13 ^AB^		99.07		0.04		0.15 ^A^	
	stocking density	20	77.03 ^A^	0.001	27.95	0.148	74.47 ^A^	<0.001	99.17	0.904	0.02	0.054	0.10	0.575
		30	72.55 ^B^		27.19		67.77 ^B^		98.70		0.01		0.12	
		40	74.45 ^B^		27.27		65.79 ^B^		98.90		0.04		0.14	
	site × stocking density			0.562		0.941		0.693		0.372		0.632		0.272
September 2018 (240th day)	site	Beihai	75.25 ^B^	<0.001	27.36 ^B^	0.031	73.02 ^C^	<0.001	95.66	0.181	0.01	0.065	0.01 ^B^	<0.001
		Qinzhou	74.40 ^B^		28.38 ^A^		79.64 ^B^		95.42		0.003		0.22 ^A^	
		Fangchenggang	77.72 ^A^		28.21 ^A^		89.41 ^A^		94.54		0.04		0.28 ^A^	
	stocking density	20	77.92 ^A^	<0.001	28.71 ^A^	<0.001	90.10 ^A^	<0.001	98.33 ^A^	0.001	0.01	0.202	0.23 ^A^	0.010
		30	74.67 ^B^		28.09 ^A^		76.70 ^B^		93.51 ^B^		0.04		0.14 ^B^	
		40	74.79 ^B^		27.15 ^B^		75.27 ^B^		93.78 ^B^		0.01		0.14 ^B^	
	site × stocking density			0.384		0.836		0.597		0.708		0.161		0.667
November 2018 (300th day)	site	Beihai	78.30 ^B^	<0.001	28.52 ^B^	<0.001	81.88 ^B^	<0.001	95.84	0.609	0.05	0.279	0.15	0.091
		Qinzhou	78.79 ^B^		28.75 ^B^		94.28 ^A^		97.18		0.07		0.23	
		Fangchenggang	82.27 ^A^		30.88 ^A^		98.41 ^A^		95.76		0.08		0.17	
	stocking density	20	82.94 ^A^	<0.001	30.94 ^A^	<0.001	100.84 ^A^	<0.001	96.67	0.550	0.08	0.060	0.19	0.233
		30	78.99 ^B^		28.37 ^B^		90.26 ^B^		96.67		0.07		0.21	
		40	77.42 ^C^		29.38 ^B^		83.49 ^C^		95.44		0.04		0.15	
	site × stocking density			0.256		0.001		0.114		0.914		0.309		0.946

Values not sharing a common superscript are significantly different (*p* < 0.05). SR (survival rate, %) = 100 × final oyster number/initial oyster number. DHG (daily shell height gain, mm/d) = 100 × (final shell height—initial shell height)/days. DMG (daily body mass gain, g/d) = 100 × (final whole weight—initial whole weight)/days.

**Table 3 biology-13-00297-t003:** Effect of stocking density on growth traits of oyster “South China No. 1” and *C. hongkongensis* conducted for 360 days in Beihai.

Treatments	Items			
	Shell Height (mm)	Shell Length (mm)	Shell Width (mm)	Total Weight (g)
Control group/20	88.06 ± 9.48 ^b^	66.29 ± 7.91 ^b^	30.39 ± 4.41 ^ab^	111.21 ± 27.86 ^a^
Control group/30	83.27 ± 9.92 ^c^	63.75 ± 8.14 ^cd^	27.93 ± 3.93 ^d^	96.68 ± 22.18 ^bc^
Control group/40	83.67 ± 9.02 ^c^	61.95 ± 7.75 ^de^	28.77 ± 5.01 ^cd^	87.51 ± 20.49 ^d^
“South China No.1”/20	96.15 ± 10.9 ^a^	68.65 ± 8.45 ^a^	31.59 ± 4.40 ^a^	117.50 ± 25.42 ^a^
“South China No.1”/30	90.49 ± 11.29 ^b^	65.71 ± 7.85 ^bc^	29.94 ± 4.92 ^bc^	100.50 ± 24.11 ^b^
“South China No.1”/40	84.61 ± 9.20 ^c^	61.13 ± 8.00 ^e^	28.78 ± 4.77 ^cd^	89.88 ± 21.83 ^cd^
One-way ANOVA (by SPSS)				
ANOVA (*P*)	<0.001	<0.001	<0.001	<0.001
Linear trend (*P*)	0.056	0.063	0.850	0.002
Quadratic trend (*P*)	0.002	0.058	0.435	0.492
Two-way ANOVA (by SPSS)				
Means of stocking density				
20	92.10 ^A^	67.47 ^A^	30.99 ^A^	114.36 ^A^
30	86.88 ^B^	64.73 ^B^	28.94 ^B^	98.59 ^B^
40	84.14 ^C^	61.54 ^C^	28.77 ^C^	88.71 ^C^
Means of breed				
Control group	85.00 ^Y^	63.99	29.03 ^Y^	98.81 ^Y^
“South China No.1”	90.42 ^X^	65.17	30.10 ^X^	102.63 ^X^
*p*-value				
Stocking density	<0.001	<0.001	<0.001	<0.001
Breed	<0.001	0.098	0.006	0.046
Stocking density × breed	0.006	0.145	0.174	0.306

Values not sharing a common superscript are significantly different (*p* < 0.05). Means in the same column with different superscripts (a, b, c for stocking density and breed treatment; A, B, C for stocking density; and X, Y for breed) are significantly different (*p* < 0.05).

**Table 4 biology-13-00297-t004:** Effect of stocking density on growth traits of oyster “South China No. 1” and *C. hongkongensis conducted* for 360 days in Qinzhou.

Treatments	Items			
	Shell Height (mm)	Shell Length (mm)	Shell Width (mm)	Total Weight (g)
Control group/20	85.37 ± 9.72 ^bc^	68.68 ± 9.00 ^b^	30.87 ± 7.00 ^b^	117.05 ± 30.39 ^a^
Control group/30	82.54 ± 9.65 ^c^	63.83 ± 8.19 ^c^	28.95 ± 4.91 ^c^	103.35 ± 30.74 ^bc^
Control group/40	82.40 ± 8.22 ^c^	62.77 ± 7.83 ^c^	30.43 ± 6.41 ^bc^	99.35 ± 23.28 ^c^
“South China No.1”/20	91.04 ± 10.71 ^a^	71.31 ± 8.11 ^a^	33.97 ± 4.64 ^a^	123.33 ± 26.03 ^a^
“South China No.1”/30	87.46 ± 10.73 ^b^	67.80 ± 7.38 ^b^	33.18 ± 5.52 ^a^	108.92 ± 28.54 ^b^
“South China No.1”/40	83.76 ± 10.03 ^c^	63.56 ± 8.78 ^c^	30.57 ± 4.27 ^bc^	96.01 ± 24.58 ^c^
One-way ANOVA (by SPSS)				
ANOVA (*P*)	<0.001	<0.001	<0.001	<0.001
Linear trend (*P*)	0.078	0.479	0.003	0.008
Quadratic trend (*P*)	0.058	0.398	0.020	0.155
Two-way ANOVA (by SPSS)				
Means of stocking density				
20	88.20 ^A^	70.00 ^A^	32.42 ^A^	120.19 ^A^
30	85.00 ^B^	65.82 ^B^	31.07 ^B^	106.14 ^B^
40	83.08 ^B^	63.17 ^C^	30.50 ^B^	97.68 ^C^
Means of breed				
Control group	83.43 ^Y^	65.09 ^Y^	30.08 ^Y^	106.58
“South China No.1”	87.42 ^X^	67.56 ^X^	32.58 ^X^	109.42
*p*-value				
Stocking density	<0.001	<0.001	0.002	<0.001
Breed	<0.001	0.001	<0.001	0.205
Stocking density × breed	0.088	0.222	0.001	0.097

Values not sharing a common superscript are significantly different (*p* < 0.05). Means in the same column with different superscripts (a, b, c for stocking density and breed treatment; A, B, C for stocking density; and X, Y for breed) are significantly different (*p* < 0.05).

**Table 5 biology-13-00297-t005:** Effect of stocking density on growth traits of oyster “South China No. 1” and *C. hongkongensis* conducted for 360 days in Fangchenggang.

Treatments	Items			
	Shell Height (mm)	Shell Length (mm)	Shell Width (mm)	Total Weight (g)
Control group/20	80.11 ± 15.66 ^b^	53.88 ± 10.50 ^c^	30.00 ± 9.46 ^b^	94.49 ± 39.96 ^b^
Control group/30	82.13 ± 11.08 ^b^	55.35 ± 9.31 ^bc^	31.81 ± 8.04 ^bc^	99.53 ± 32.23 ^b^
Control group/40	80.51 ± 11.85 ^b^	54.09 ± 9.59 ^c^	31.19 ± 10.55 ^b^	94.74 ± 31.43 ^b^
“South China No.1”/20	86.91 ± 13.77 ^a^	59.97 ± 8.64 ^a^	34.09 ± 58.87 ^a^	113.15 ± 36.60 ^a^
“South China No.1”/30	81.61 ± 9.76 ^b^	58.08 ± 10.00 ^ab^	34.09 ± 6.27 ^a^	97.25 ± 24.21 ^b^
“South China No.1”/40	84.09 ± 12.71 ^ab^	57.72 ± 10.54 ^ab^	33.64 ± 5.94 ^a^	101.15 ± 27.75 ^b^
One-way ANOVA (by SPSS)				
ANOVA (*P*)	0.003	<0.001	0.001	0.001
Linear trend (*P*)	0.026	<0.001	<0.001	0.118
Quadratic trend (*P*)	0.306	0.218	0.248	0.111
Two-way ANOVA (by SPSS)				
Means of stocking density				
20	83.30	56.83	31.98	103.31
30	81.94	56.67	32.91	98.66
40	82.52	55. 95	32.47	97.90
Means of breed				
Control group	80.92 ^Y^	54.44 ^Y^	31.00 ^Y^	96.25 ^Y^
“South China No.1”	84.13 ^X^	58.53 ^X^	33.90 ^X^	103.67 ^X^
*p*-value				
Stocking density	0.586	0.653	0.253	0.725
Breed	0.009	<0.001	<0.001	0.001
Stocking density × breed	0.149	0.200	0.259	0.012

Values not sharing a common superscript are significantly different (*p* < 0.05). Means in the same column with different superscripts (a, b, c for stocking density and breed treatment; A, B, C for stocking density; and X, Y for breed) are significantly different (*p* < 0.05).

**Table 6 biology-13-00297-t006:** Dominant phytoplankton in Beihai, Qinzhou and Fangchenggang.

Site	Species	Average Dominance	Average Frequency of Occurrence	Phytoplankton Species Abundance (Cells·L^−1^)	Sum of Phytoplankton Species Abundance (Cells·L^−1^)
Beihai	*Skeletonema*	0.70	100	4193	5928
	*Pleurosigma*	0.03	100	159	
	*Eucampia*	0.03	71.43	197	
	*Navicula*	0.03	100	152	
	*Ditylum*	0.00	28.57	28	
	*Chaetoceros*	0.04	57.14	337	
	*Rhizosolenia*	0.02	85.71	166	
	*Thalassionema*	0.02	71.43	174	
	*Coscinodiscus*	0.02	85.71	136	
	*Synedra*	0.00	14.29	23	
	*Nitzschia*	0.00	28.57	16	
	*Triceratium*	0.00	14.29	1	
	*Bacillaria*	0.01	71.43	61	
	*Licmophora*	0.00	57.14	47	
	*Cerataulina*	0.00	14.29	19	
	*Thalassiosira*	0.00	28.57	60	
	*Lauderia*	0.00	14.29	35	
	*Meuniera*	0.01	57.14	124	
Qinzhou	*Skeletonema*	0.68	100	4589	6466
	*Rhizosolenia*	0.01	42.86	84	
	*Ditylum*	0.03	57.14	212	
	*Eucampia*	0.01	28.57	203	
	*Chaetoceros*	0.00	28.57	65	
	*Thalassionema*	0.02	71.43	229	
	*Navicula*	0.02	100	152	
	*Nitzschia*	0.00	57.14	15	
	*Pleurosigma*	0.02	100.00	158	
	*Coscinodiscus*	0.05	85.71	290	
	*Synedra*	0.00	14.29	4	
	*Cerataulina*	0.00	14.29	7	
	*Bacillaria*	0.01	71.43	81	
	*Thalassiosira*	0.01	57.14	91	
	*Hemiaulus*	0.00	28.57	34	
	*Biddulphia*	0.00	28.57	52	
	*Lauderia*	0.00	14.29	26	
	*Licmophora*	0.00	28.57	16	
	*Schroderella*	0.00	14.29	121	
	*Odontella*	0.00	14.29	15	
	*Detonula*	0.00	14.29	22	
Fangchenggang	*Skeletonema*	0.68	100	6343	9118
	*Thalassionema*	0.03	85.71	255	
	*Pleurosigma*	0.02	100	147	
	*Navicula*	0.02	100	170	
	*Ditylum*	0.00	57.14	30	
	*Coscinodiscus*	0.02	85.71	176	
	*Chaetoceros*	0.06	71.43	667	
	*Rhizosolenia*	0.01	57.14	90	
	*Nitzschia*	0.00	42.86	13	
	*Bacillaria*	0.00	28.57	26	
	*Triceratium*	0.00	14.29	3	
	*Thalassiosira*	0.02	71.43	374	
	*Synedra*	0.00	14.29	11	
	*Lauderia*	0.01	57.14	196	
	*Eucampia*	0.01	28.57	312	
	*Licmophora*	0.00	42.86	26	
	*Biddulphia*	0.00	14.29	7	
	*Meuniera*	0.00	28.57	99	
	*Guinardia*	0.00	28.57	150	
	*Odontella*	0.00	28.57	23	

**Table 7 biology-13-00297-t007:** Genetic diversity parameters of seven oyster populations.

Loci	Parameter	Population						
		BHC	BHN	QZC	QZN	FCC	FCN	QDT
Ch311	*N_A_*	5.00	5.00	4.00	4.00	5.00	5.00	5.00
	*N_E_*	1.52	1.89	1.78	2.56	2.04	1.89	2.68
	*H_O_*	0.28	0.36	0.64	0.89	0.42	0.56	0.89
	*H_E_*	0.35	0.63	0.49	0.62	0.52	0.52	0.64
	*P*	0.00 *	0.00 *	1.00	1.00	0.00 *	0.33	1.00
Ch319	*N_A_*	4.00	5.00	3.00	4.00	4.00	5.00	3.00
	*N_E_*	2.70	2.03	2.06	2.31	2.88	2.37	2.05
	*H_O_*	0.92	0.42	1.00	0.53	1.00	1.00	0.94
	*H_E_*	0.64	0.51	0.52	0.58	0.66	0.59	0.52
	*P*	1.00	0.00 *	1.00	0.06	1.00	1.00	1.00
Ch423	*N_A_*	8.00	5.00	8.00	4.00	8.00	4.00	5.00
	*N_E_*	2.94	2.24	3.68	2.12	3.52	2.01	2.88
	*H_O_*	0.72	0.72	0.58	0.69	0.81	0.78	0.69
	*H_E_*	0.69	0.56	0.74	0.54	0.75	0.51	0.66
	*P*	0.67	0.72	0.02	0.10	0.19	1.00	0.03
Ch417	*N_A_*	4.00	6.00	7.00	6.00	4.00	6.00	6.00
	*N_E_*	1.96	2.08	3.84	1.94	2.23	3.09	3.30
	*H_O_*	0.58	0.67	0.58	0.61	0.75	0.78	0.78
	*H_E_*	0.50	0.53	0.75	0.49	0.56	0.69	0.71
	*P*	0.86	1.00	0.03	1.00	0.99	0.90	0.01 *
Ch405	*N_A_*	5.00	4.00	5.00	8.00	6.00	9.00	6.00
	*N_E_*	3.61	2.10	2.89	4.29	3.47	7.12	2.20
	*H_O_*	0.75	0.72	0.50	0.67	0.92	0.61	0.67
	*H_E_*	0.73	0.53	0.66	0.80	0.72	0.87	0.55
	*P*	0.58	1.00	0.05	0.00 *	0.53	0.00 *	1.00
Ch697	*N_A_*	5.00	8.00	6.00	6.00	6.00	8.00	9.00
	*N_E_*	1.87	4.82	2.51	2.60	2.86	3.72	4.73
	*H_O_*	0.58	0.83	0.39	0.64	0.83	0.81	0.75
	*H_E_*	0.47	0.80	0.61	0.62	0.66	0.74	0.80
	*P*	1.00	0.25	0.02	0.11	1.00	0.78	0.08
Ch307	*N_A_*	4.00	4.00	3.00	5.00	5.00	5.00	4.00
	*N_E_*	2.91	2.07	1.18	2.36	2.94	2.57	1.87
	*H_O_*	0.75	0.64	1.00	0.92	0.89	0.94	0.64
	*H_E_*	0.67	0.57	0.58	0.60	0.68	0.62	0.47
	*P*	0.74	0.04	1.00	1.00	0.98	1.00	1.00
Ch317	*N_A_*	3.00	4.00	4.00	5.00	5.00	2.00	4.00
	*N_E_*	1.18	1.41	1.3	1.60	2.63	1.15	1.33
	*H_O_*	0.17	0.39	0.19	0.50	0.86	0.14	0.28
	*H_E_*	0.16	0.40	0.28	0.48	0.67	0.13	0.25
	*P*	1.00	0.01 *	0.00 *	0.02	0.15	1.00	1.00
Ch411	*N_A_*	6.00	5.00	9.00	5.00	7.00	5.00	4.00
	*N_E_*	2.24	3.96	4.60	1.73	3.10	1.81	2.49
	*H_O_*	0.67	1.00	0.67	0.36	0.92	0.53	0.83
	*H_E_*	0.56	0.76	0.79	0.43	0.69	0.46	0.61
	*P*	0.10	1.00	0.02	0.03	1.00	0.97	1.00
Ch414	*N_A_*	6.00	5.00	7.00	2.00	7.00	8.00	6.00
	*N_E_*	3.78	1.66	2.78	1.39	4.69	3.02	2.92
	*H_O_*	0.92	0.47	0.81	0.33	0.86	0.64	0.47
	*H_E_*	0.76	0.40	0.65	0.28	0.82	0.68	0.67
	*P*	0.10	1.00	1.00	1.00	0.43	0.62	0.00 *
Mean	*N_A_*	5.00	5.10	5.60	4.90	5.70	5.70	5.20
	*N_E_*	2.47	2.43	2.67	2.29	3.03	2.87	2.65
	*H_O_*	0.63	0.62	0.64	0.61	0.83	0.68	0.69
	*H_E_*	0.55	0.57	0.61	0.54	0.67	0.58	0.59

BHC = *C. hongkongensis* at Beihai; BHN = oyster “South China No. 1” at Beihai; QZC = *C. hongkongensis* at Qinzhou; QZN = oyster “South China No. 1” at Qinzhou; FCC = *C. hongkongensis* at Fangchenggang; FCN = oyster “South China No. 1” at Fangchenggang; QDT = *C. gigas* from Qingdao; *N_A_* = the allele number; *N_E_* = effective allele number; *H_o_* = observed heterozygosity; *H_E_* = expected heterozygosity. * indicates significant departure from the Hardy–Weinberg equilibrium after Bonferroni correction (*p* < 0.05).

## Data Availability

Data will be made available on request.

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
