# Peer review of "Aquaculture Performance and Genetic Diversity of a New [(Crassostrea hongkongensis ♀ × C. gigas ♂) ♂ × C. hongkongensis ♀] Variety of the Oyster “South China No. 1” in Beibu Gulf, China"

_biology, 2024, doi:10.3390/biology13050297_

Round 1
Reviewer 1 Report
Comments and Suggestions for Authors
attached in review report

A through revision in terms of grammar, phrases is highly required
Author Response
1.Limited Context: The report lacks background information on "South China No.1," including its origin, genetic makeup, and the significance of its backcross advantage.
Response: Thanks so much for your reasonable suggestion. We have replenished the background information of "South China No.1" in the introduction. Please see the detail in lines 74-89.
2.Unclear Objectives: The research goals are not clearly defined. The text mentions exploring stocking density effects on "South China No.1" and comparing it to Crassostrea hongkongensis, but specific research questions or hypotheses are absent.
Response: Thank you for the reviewer’s comment. This study aims to explore the effects of stocking density on the growth performance of “South China No.1”, compared their growth performance and genetic diversity to Crassostrea hongkongensis, and found the best place breeding site for “South China No.1” in Beibu Gulf.
3.Opaque Methodology: The methods section is inadequate. Crucial details regarding sampling techniques, experimental design, statistical analyses, and genetic analysis methods are missing, hindering evaluation of the study's rigor.
Response: Thank you very much for your good question. The experimental oysters were separately kept in lanterns (Figure 1). From top to bottom, the lantern of A was breeded stocking density of 20, 20, 30, 30, 40 and 40 oysters/substrate, respectively. From top to bottom, the lantern of B was breeded stocking density of 30, 30, 40, 40, 20 and 20 oysters/substrate, respectively. From top to bottom, the lantern of C was breeded stocking density of 40, 40, 20, 20, 30 and 30 oysters/substrate, respectively. We used two-factor Analysis of variance Variance (ANOVA) to establish the effects of location, stocking density and their interaction on growth of “South China No.1”, one-way Analysis of Variance followed byand Duncan's multiple comparison test to compare the growth parameters of the two breeds (“South China No.1” and Crassostrea hongkongensis ) with different stocking densities. Genetic analysis was calculated by Popgen 32.0 and by Gervus.

Figure 1. Estrutura de balsa e lanternas utilizadas no cultivo de ostras [1].
[1] de Oliveira Cardoso Junior Leônidas.; Lavander, HD.; Neto, SRDS.; de Souza, AB.; Silva, LOBD.; Gálvez, AO. Crescimento da ostra crassostrea rhizophorae cultivada em diferentes densidades de estocagem no litoral norte de pernambuco. Pesquisa Agropecuária Pernambucana 2012, 17, 10-14.
4.Insufficient Discussion: While results on growth performance and genetic diversity are presented, a deeper interpretation and discussion are lacking. The report doesn't explore potential mechanisms behind the observed effects or connect the findings to broader issues in oyster aquaculture and genetic conservation.
Response: Thank you for reminding us for this. We have supplemented the discussion. Please see the detail in the part of disscussion.
5.Inconsistent Presentation: Results are presented inconsistently, with some findings descriptive and others supported by statistical analyses. Consistent use of statistical measures (p-values, effect sizes) is necessary.
Response: We thank the referee for his/her kind suggestion about this. For phytoplankton statistics we used descriptive analysis. The purpose of phytoplankton statistics is to understand the main phytoplankton in the local sea area and assess their species and quantity as one of the directions for analyzing the food sources of oysters. Statistical analysis of algal abundance was referenced by Huang et al [2].
[2] Huang, Yuan.; Cen, Jingyi.; Liang, Qianyan.; Lyu, Songhui.; Wang, Jianyan. Study on the community structure of eukaryotic phytoplankton in Shenzhen Bay based on high-throughput sequencing technology. Journal of Tropical Oceanography 2023, 43.
6.Superficial Comparison: The comparison between "South China No.1" and Crassostrea hongkongensis is shallow. Deeper analysis with statistical comparisons and discussion of both similarities and differences would strengthen the conclusions.
Response: This is a good suggestion. We have reinforced the discussion of "South China No.1" and Crassostrea hongkongensis. Please see the detail in lines 324-357.
7.Missing Conclusion: The report lacks a concluding section that summarizes the key findings, their implications, and potential future research directions or practical applications in oyster aquaculture and genetic conservation. Therefore I recommend its major revisions and suggest for resubmission after incorporation of the comments.
Response: Thank you for reminding us for this. We have revamped the conclusion. Please see the detail in lines 432-443.

Reviewer 2 Report
Comments and Suggestions for Authors
At first, The manuscript does not include any tables and figures and hence, I cannot understand the research outcomes itself. Second, the introduction does not show the justification and novelty of the research. The authors mentioned “There have been many reports on the genetic diversity of artificially selected populations of some economic shellfish, such as C. gigas[19], Crassostrea sikamea [20] and Crassostrea virginica [21] in L68-70. In that case, this research was just repeating using different species. There are also many studies in stocking density in a number of oyster species. Therefore, the two research purposes are no longer novel. “South China No.1” is used in the title and abstract. However, the term is not scientific names, not commonly used and unfamiliar and hence, the authors should not use the title and abstract. Authors also are needed to define the term with citation in the introduction and then they should start to use. I do not see any citation how the authors can use “South China No.1”. Does it just define by authors? If so, the term is invalid. The font sizes are inconsistent in the manuscript. “Beibu gulf” should be “Beibu Gulf” throughout the manuscript. The proofreading is definitely needed before submission. Since the manuscript is incomplete without tables and figures, I do not suggest anything about results, discussion and conclusion at this stage.
Comments on the Quality of English LanguageProofreading should be needed.
Author Response
- At first, The manuscript does not include any tables and figures and hence, I cannot understand the research outcomes itself.
Response: Thanks so much for your reasonable suggestion. We have incorporated the tables and figures directly into the manuscript.
- The authors mentioned “There have been many reports on the genetic diversity of artificially selected populations of some economic shellfish, such as C. gigas[19], Crassostrea sikamea [20] and Crassostrea virginica [21]in L68-70.In that case, this research was just repeating using different species. There are also many studies in stocking density in a number of oyster species. Therefore, the two research purposes are no longer novel.
Response: Thank you for the reviewer’s comment. In Crassostrea hongkongensis culture industry, there are often bad phenomena such as poor disease resistance, stress resistance, individual miniaturization and unit yield reduction. The total weight and yield of a new [(Crassostrea hongkongensis ♀ × Crassostrea gigas variety ♂) ♂ × Crassostrea hongkongensis ♀] variety “South China No.1” is 17.1% and 23.1% higher than that of ordinary Crassostrea hongkongensis. This study can be used as a guide to further improve “South China No.1” aquaculture practices, and provide scientific reference for the introduction and popularization in Beibu Gulf of Guangxi.
3.“South China No.1” is used in the title and abstract. However, the term is not scientific names, not commonly used and unfamiliar and hence, the authors should not use the title and abstract. Authors also are needed to define the term with citation in the introduction and then they should start to use. I do not see any citation how the authors can use “South China No.1”. Does it just define by authors? If so, the term is invalid.
Response: Thank you very much for your good question. We have supplemented the source of “South China No. 1” in title. “South China No. 1” is a new variety of Crassostrea hongkongensis and Crassostrea gigas [1]. We have cited in L79-84. Please see the detail in introduction section.
[1] Yu, ZN.; Zhang, YH.; Zhang, Y.; Wang, ZP.; Xiao, S.; Li, J.; Xiang, ZM.; Ma, HT. New aquaculture oyster "South China No. 1". China Fisheries 2017, 2, 86-89.
4.The font sizes are inconsistent in the manuscript. “Beibu gulf” should be “Beibu Gulf” throughout the manuscript. The proofreading is definitely needed before submission.
Response: We thank the referee for his/her kind suggestion about this. We have corrected spelling of “Beibu gulf”. Please see the detail in lines 40,52 and 56.

Reviewer 3 Report
Comments and Suggestions for Authors
Dear authors,
The study of a new variety of oyster called “South China No.1” in the Beibu Gulf is interesting from an aquaculture point of view.
Your work is solid, the entire manuscript is coherent, the methodology is well described and the discussion seems very good to me (perhaps the best part of the manuscript). It addresses all points of the work with current references and from very reliable sources.
My review of the manuscript is favorable. However, there are some spelling errors in the manuscript that need to be revised. Additionally, in line 335, the correct word is 'Bonferroni.' Finally, the format of the tables would also need to be adapted. These are minor errors that are not considered significant.
Author Response
The study of a new variety of oyster called “South China No.1” in the Beibu Gulf is interesting from an aquaculture point of view.
Your work is solid, the entire manuscript is coherent, the methodology is well described and the discussion seems very good to me (perhaps the best part of the manuscript). It addresses all points of the work with current references and from very reliable sources.
My review of the manuscript is favorable. However, there are some spelling errors in the manuscript that need to be revised. Additionally, in line 335, the correct word is 'Bonferroni.' Finally, the format of the tables would also need to be adapted. These are minor errors that are not considered significant.
Response: Thanks so much for your reasonable suggestion. We have replaced “Bonfreni” with “Bonferroni” in the line 425.

Round 2
Reviewer 2 Report
Comments and Suggestions for Authors
Thanks for your revision. I am happy to see the revised manuscript and the complete response to my inquires and concerns.
Comments on the Quality of English LanguageMinor editing of English language will be required.